# Effect of the Foresight Horizon on Computation Time and Results Using a Regional Energy Systems Optimization Model

Jessica Thomsen *, Noha Saad Hussein, Arnold Dolderer and Christoph Kost

Fraunhofer Institute for Solar Energy Systems, Heidenhofstr. 2, 79110 Freiburg, Germany;
noha.saad.hussein@ise.fraunhofer.de (N.S.H.); arnold.dolderer@ise.fraunhofer.de (A.D.);
Christoph.Kost@ise.fraunhofer.de (C.K.)
* Correspondence: jessica.thomsen@ise.fraunhofer.de

**Abstract:** Due to the high complexity of detailed sector-coupling models, a perfect foresight optimization approach reaches complexity levels that either requires a reduction of covered time-steps or very long run-times. To mitigate these issues, a myopic approach with limited foresight can be used. This paper examines the influence of the foresight horizon on local energy systems using the model DISTRICT. DISTRICT is characterized by its intersectoral approach to a regionally bound energy system with a connection to the superior electricity grid level. It is shown that with the advantage of a significantly reduced run-time, a limited foresight yields fairly similar results when the input parameters show a stable development. With unexpected, shock-like events, limited foresight shows more realistic results since it cannot foresee the sudden parameter changes. In general, the limited foresight approach tends to invest into generation technologies with low variable cost and avoids investing into demand reduction or efficiency with high upfront costs as it cannot compute the benefits over the time span necessary for full cost recovery. These aspects should be considered when choosing the foresight horizon.

**Keywords:** optimization; energy system model; myopic; perfect foresight

## 1. Introduction

Energy system optimization models are widely used tools to investigate questions concerning the energy sector, its potential developments, effects of new technologies, or price developments to name a few. With increasing efforts to transform the energy systems in order to face and mitigate climate change, planning horizons, solution space and exchange rate of technologies increase in many energy-system models. Additionally, the power sector has a high complexity due to its network-based transmission and distribution. On the one hand, modeling always requires certain computational resources. On the other hand, researchers as well as energy system planners work with simplifications and idealized approaches. This boils down to a delicate balance between reducing the complexity and hence ensure solvability as well as a reasonable run-time with the available technical resources and the amount of detail required to actually generate a proper evaluation and assessment of energy system questions. Model runs often reach computational limits so that system planners reach out to solutions like reducing the number of covered time-steps, aggregating the number of considered nodes to solve energy system problems within a reasonable run-time. Future carbon-neutral energy systems are not only characterised by high shares of renewable energy, but also by an increasing coupling of energy sectors. When adding sector coupling to the problem the scope is not only doubled but could lead to an exponential increase of the variables within the mathematical problem. This requires different approaches that lead to a reduction of the model run-time according to the available computational resources.

When considering energy system optimization models that forecast future periods and investment decisions, such as [1–11] amongst others, two approaches can be found. Many

models use a perfect foresight method over the whole optimization period [1,2], meaning that all the given information and developments are known to the solver from the starting point. Usually, several years are considered in such an analysis, and as a result the number of time-steps per year is reduced sharply to be able to solve the mathematical problem in reasonable time. The second approach is a myopic optimization, where the optimization period is broken down into sub-periods that are optimized consecutively [1,12]. Each of the approaches has its strengths and weaknesses. For example, in reality, no stakeholder has a perfect foresight of the future, so the perfect foresight approach always renders the best possible solution in an ideal world. However, the limited foresight in the myopic approach neglects that stakeholders have a certain expectation of future developments when they evaluate their investment decisions. Nonetheless, the myopic approach needs considerably less computation time and can thus speed up the modeling phase.

Some undertaken evaluations showed that for power system models with consistent optimization constraints, the results are similar with both approaches unless parameters include a "shock", which in reality would be an unexpected development that denotes a stark shift away from the past development. [13] However, past evaluations have mainly looked at national energy system models and mostly the power sector. Therefore, the authors identified a knowledge gap in the local energy systems field as well as the sector coupling topic and whether a myopic approach can be suitable in certain scenario settings. Hence, this paper investigates under which criteria one can benefit from the myopic advantages like shorter run times and under which settings perfect foresight should be preferred despite its longer runtime. The analysis is conducted with the regional sector-coupling energy system model DISTRICT presented in [14–17]. From this, a guideline that helps in choosing the more appropriate approach is derived. The remainder of the paper is structured as followed: Section 2 gives a brief overview of relevant literature and Section 3 gives an overview on the model DISTRICT and the two expansion approaches. Section 4 describes the analyzed system, necessary assumptions, and the scenario design. In Section 5, the scenario results are compared for both expansion approaches, concluding with a discussion on the applicability of each method. Section 6 summarizes the recommended applicability of each method depending on the research question at hand.

## 2. Literature Review

As a vast amount of literature has been published looking into energy system models at national as well as at distributed level, this literature review focuses on work especially dedicated to the influence of the time horizon and time steps in optimization models.

Ref. [3] provides an review on existing models on local or district levels. It summarizes the different advantages and disadavantes, mainly connected with data requirements, physical robustness, accuracy, runtime, applicability, and specific use. Due to the results in this paper, the cirmcumstances or research question highly interact with the chosen model approach.

Ref. [1] tested perfect forecast optimization, time step approach and stochastic optimization on different research questions. In the paper, the advantages of each approach is quantitatively assessed. Shocks could be better analyzed with the time step approach, whereas stoachstic optimizations shows advantages on different energy price scenarios. Similar to this paper, [2] also highlights the aspects of flexibility and lost opportunities in the case of a shock if modeled with a time step approach. In general, it is expected that this approach represents a more realistic reaction of the system of a real world, unexpected shock event.

Ref. [12] assess the effect of an alternative sequential decision approach instead of a perfect foresight approach within the optimization model MESSAGE. The main difference lies in the limited foresight of the sequential approach, where the optimizer only has knowledge of some of the future information. However, unlike the standard myopic approach, this solution provides the possibility to alter some of the decisions at a later stage. This is not applicable for decisions like investment decisions. This approach was tested

on three differnet scenarios and one variant. The results show that the limited foresight approach represents reality more accurately. Decision makers base a lot of their choices on the needs of the present and do not consider many of the future changes. Since the present relies heavily on fossil fuels for example, investments in fossil technolgies can be made which disregard climate constraints of the future.

The effects of limited foresight on technology adoption is the focus of [18]. One of the results show that minimizing the cost of each time period can result in a non optimal solution in total. The study also shows that with a limited foresight approach and long decision periods early adoption of new technologies is more likely. However when reaching a certain length, the influence gets minimized and becomes less visible. The authors find that "the range of technological bifurcation is much larger than that with perfect foresight, but uncertainty in technological learning tends to reduce the range by removing the early adoption paths of a new technology" [18].

Ref. [19] develops a moypic version of the perfect forecast model Perseus. It splits the optimization into multiple and individually smaller optimization problems to improve the computing time of the model. With a stable input parameter set, the new myopic model is sloved in only 10% of original computing time of the perfect foresight model.

Ref. [20] studied the development of the European electricity sector under different foresight and $CO_2$ budget approaches with the model dynELMOD. The reduced foresight leads to stranded investments compared to the perfect foresight approach. More gas-powered plants are installed, shifting generation from coal to gas, but as $CO_2$ emissions decline, so do their full load hours. Additionally, they find that using a budget approach for $CO_2$ emissions, where an aggregated budget for the whole period is given to the model, leads to a faster reduction of $CO_2$ emissions than in those scenarios with periodical $CO_2$ emission targets.

Ref. [21] examine the influence of the foresight horizon on decarbonization pathways for the power system under different strategies. Those strategies include waiting for a so-called "unicorn" technology to emerge that allows reaching carbon reduction goals easily. When waiting in vain, the myopic approach leads to considerably higher greenhouse gas emissions and installed capacities. Furthermore, even with the unicorn technology emerging, the myopic approach shows high overcapacities compared to the perfect fore-sight approach. The authors conclude that to reach the targets of reducing greenhouse gas emissions, one should try to increase the foresight of parameter development in the real world by establishing a corresponding political framework.

Ref. [13] explore decision making and investment strategies in a power sector model for Belgium under a different foresight and a myopic approach. It is shown that the myopic approach aims to maximize short-term profits. In the presented case, this equals investment into coal fired plants without carbon capture storage in the first periods, when carbon prices are low. These lead to higher carbon emissions and thus mitigating costs at the end of optimization. The authors find that the myopic approach does not extrapolate visible price trends and thus falls short to represent investments under uncertainties as in reality the trend of certain parameters can be extrapolated beyond the current period of interest. They also argue the perfect foresight approach to be unrealistic as a representation for investment decisions, as a sudden jump in prices cannot be foreseen by all investors.

Compared to the existing literature, the paper analyzes the advantages of the myopic and perfect forecast optimization for a regional sector-coupling energy system model. Furthermore, it discusses which approach might be more suitable for certain scenario settings or research includes aspects such as price shocks or demand reduction technologies with high investment cost. In the discussion section, a comparison with existing literature is undertaken.

## 3. Methodological Approach

The present methodological comparison is undertaken for the regional energy system model DISTRICT. DISTRICT is an optimization model that targets costs minimization.

### 3.1. General Modeling Approach

The model covers the electricity, heat, and cold sectors for a regional energy system. The model takes into account renewable generation technologies (such as photovoltaic, solar thermal, and wind energy), storage technologies, sector-coupling technologies (such as combined heat and power plants and heat pumps), and demand-side management at prosumer level. The regional system is subdivided into model areas to consider electricity, heat, and cold grid infrastructure between the model areas. Model areas can be scaled from single buildings up to districts by adjusting the input data. Retrofit and its accompanying reduction in heat demand are also included (technically, it would also be possible to consider refurbishment for districts, but the model formulation would require all buildings aggregated to one model area to be refurbished to the same energy efficiency standard; so far, it has only been applied to single buildings) and are optimized endogenously, considering the tradeoffs between retrofit, available technologies, and corresponding costs. Within each model area, energy can be generated, consumed directly, stored, and/or fed into the grid. Remaining energy demand can be covered by withdrawing energy from the grid if a grid connection exists. For the electricity sector, the regional energy system is embedded in the national electricity market and disposes of a connection to the superior grid. This provides the opportunity to assess potential trades at the central energy markets, marked by load flows in and out of the superior grid.

DISTRICT's objective function minimizes the total system (TC) cost. This includes the total variable cost (VC), total fixed (FC), and investment cost, which is included in the model based on an annuity calculation (AN) $CO_2$ Emission Cost and a cost variable for the buying and selling electricity and heat respectively across the system boarder.

$$Total\ Cost = Variable\ Cost + Fix\ Cost + Annuities + Trade\ Sum\ electricity + Trade\ Sum\ Heat + CO2\ Emission\ Cost \quad (1)$$

As this paper focuses on comparing expansion methods, these are presented in the following sections. For the detailed formulation of the remaining model aspects, cf. [14–17]. The model includes the possibility to switch between the perfect foresight and myopic expansion approaches for the optimization, giving the user the opportunity to select the most appropriate approach for their current research question.

### 3.2. Perfect Foresight Expansion

The perfect foresight approach is used in many energy-system models to find an optimal system development. Under perfect foresight, the model has all information about the whole optimization period at once [1,18]. Thus, the optimization is capable of adjusting decisions of the present according to future developments [1,19].

Figure 1 illustrates the perfect foresight approach within an energy system model in general. The model receives a certain set of input data that covers the whole optimization period. Depending on the case, additionally, a start system is given as a basis. The optimization is executed over the whole period at once, using all input data for all time-steps considered. As a result, the objective value and the corresponding system values are determined over the whole optimization period. The model thus is able to account for future price developments in the first time-steps already, adjusting the variables to derive the optimal target value [1].

### 3.3. Myopic Expansion

The myopic approach divides the optimization period into several shorter periods, generating several optimization problems [12]. These sub-problems are smaller and thus easier to solve than very large optimization problems. Within the myopic approach, there are two types: One where the sub-problems are aligned and one where the sub-problems overlap each other in the covered time period allowing decisions to be partially revised in the following optimization period [12]. For the model DISTRICT, the first approach is implemented and illustrated in Figure 2. In contrast to the perfect foresight approach, the

myopic recursive approach only has information about the current optimization period. Thus, it has a limited foresight of parameter developments and can only make decisions based on the information for the sub-period that it currently optimizes [1]. In consequence, future developments of parameters cannot be foreseen, preventing the optimization in earlier periods to adjust its values to future developments such as price increases.

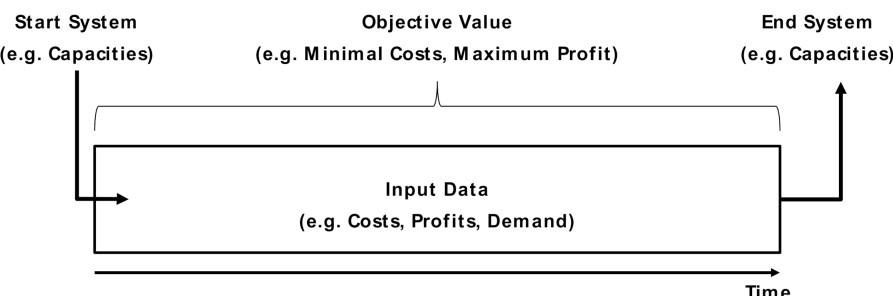

**Figure 1.** Illustration of the perfect foresight approach.

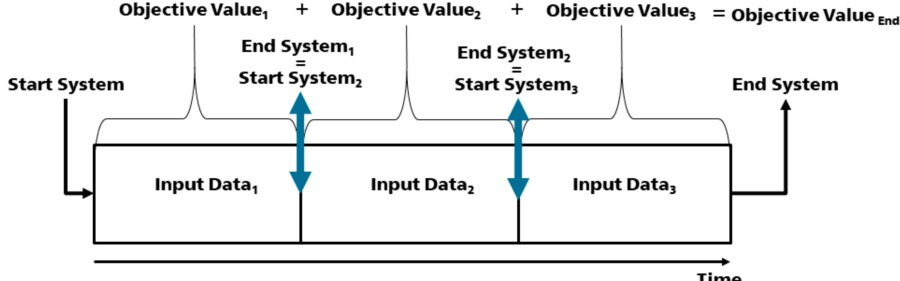

**Figure 2.** Illustration of the myopic recursive approach, where the optimization problem is divided into several smaller optimization problems that are optimized consecutively without information about future periods.

In the literature, various terms exist for the myopic approach, among others, time-step [1] or recursive-dynamic approach [22]. These notations can lead to different interpretations of the method. For example, all energy system models have a time horizon that is separated into different time-steps, which is not related to the chosen expansion approach. Furthermore, the myopic approach does not use a real recursion by definition, rather it splits the complexity of the problem and solves the sub-problems sequentially [1], as shown in Figure 2.

Generally, the myopic approach also disposes of a perfect foresight within each optimization period, having all the information for this particular sub-period. Hence, the concept of a perfect foresight is also partially used in the myopic approach. Due to this, the term "perfect foresight" can lead to confusion. Nevertheless, in the literature, it is the dominant term for the aforementioned expansion approach and will thus be used accordingly in the publication at hand.

## 4. Case Study Design

The analyzed system represents part of the Weingarten district in Freiburg im Breisgau, Germany. It consists of six regions, each representing one building.

### 4.1. System Layout

The buildings consist of five apartment buildings and one row housing block (ID 4). The latter includes eight apartments with three to four inhabitants each (Table 1). The five apartment buildings (ID 1, 2, 3, 5, and 6) include 30 to 120 apartments per building. On average, there are two persons per apartment. The KfW (Kreditanstalt für Wiederaufbau)

standards describe the energy reduction from 100%, being the current standard, to 85% or 55%, respectively.

**Table 1.** Overview and key data of buildings included in the case study (check Appendix A Figure A1 for more details).

| Building | Living Area [m²] | Current Energy Demand [MWh$_{th}$/a] | KfW 100 [MWh$_{th}$/a] | KfW 85 [MWh$_{th}$/a] | KfW 55 [MWh$_{th}$/a] | Energy Demand for Hot Water [MWh$_{th}$/a] |
|---|---|---|---|---|---|---|
| 1 | 2220.4 | 283.8 | 94.7 | 72.5 | 30.2 | 64.2 |
| 2 | 1871.0 | 229.3 | 69.2 | 54.7 | 16.3 | 60.5 |
| 3 | 7929.6 | 658.4 | 229.1 | 162.5 | 34.6 | 235.7 |
| 4 | 856.8 | 111.1 | 41.8 | 32.4 | 14.9 | 20.5 |
| 5 | 2435.1 | 232.0 | 72.9 | 48.3 | 12.3 | 96.4 |
| 6 | 2106.7 | 198.6 | 56.3 | 40.8 | 7.8 | 79.8 |

The corresponding electricity grid is derived from the existing grid installed in the area; see Figure 3. It represents a low voltage grid and a transformer, which is located in the node connected to the superior grid level, where transactions with the electricity spot market are accounted for. The transformer capacity is fixed based on the analyzed buildings' demand to 340 kW. Each grid connection has a capacity of 166 kW. Unlike the electrical grid, the district heating grid, (Figure 3, right), is assumed. The node in the west (left) of the system, which, is called "central region" for the heating system and represents the transformer, connecting the electrical grid to the superior grid, in the electrical system. At this central location, large-capacity technologies such as central CHP plants or central storage systems can be installed to feed into the corresponding grid.

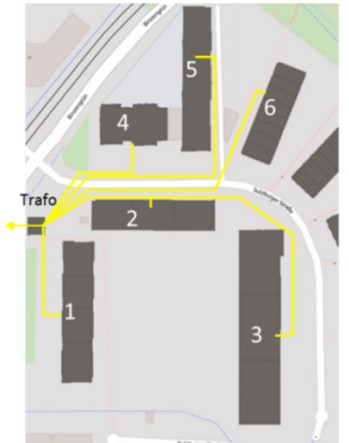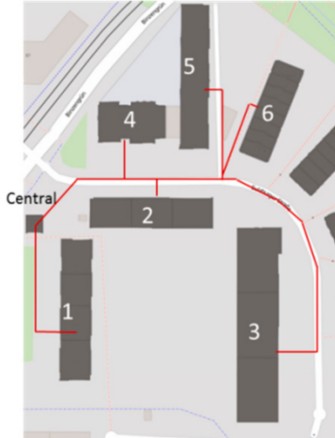

**Figure 3.** Model region including electricity grid, demand regions, and spot market connection area (**left**) and heating grid, demand regions, and central region (**right**).

Each building has an individual heating and electrical demand. The heating demand temperature is dependent on the building standard. In the case of a refurbished building, the required space heating is in low temperature. In contrast, a non-refurbished 'old' building has high-temperature heating demand.

These demands can be covered either by existing technologies installed in the buildings or by newly deployed technologies. It is assumed that in four of the six buildings, gas boilers are installed. In buildings 1 and 4, the boilers will be decommissioned in 2029 and a new installation is required in 2030. In buildings 5 and 6, the gas boilers will be decommissioned earlier and require a replacement technology in 2025. This set up forces the system to install new capacities throughout the optimization period and, therefore, helps to analyze and evaluate system and technology effects over time caused by different circumstances.

### 4.2. Potential Generation Technologies

There are similar technology potentials in each model region and hence each building. The exact potentials are determined as follows: For non-retrofitted buildings, potentials are set to 110% of the current peak demand, i.e., they exceed peak demand by 10%. This applies to CHP, heat pumps, and boilers. As heat pumps cannot provide high-temperature heat, their potential is limited to 110% of the peak demand of the KfW 100 standard.

PV and solar thermal potentials are limited to 80% of the roof areas of each building since 20% are reserved for other uses. DISTRICT also considers that areas used for PV cannot be used for solar thermal installations, i.e., if 50% of the roof are used for PV installations, only the remaining 30% is available for the installation of solar thermal panels. The heating demand of each building has a specific temperature level (high, low, or hot water) and the chosen mix of supply technologies must be able to meet the demand at all corresponding levels. Table 2 displays the considered generation technologies and their corresponding energy type and temperature levels.

**Table 2.** Energy generation technologies and provided energy type and temperature levels.

| Energy Generation Technologies | High Temperature Heat | Low Temperature Heat | Hot Water | Electricity |
|---|---|---|---|---|
| Gas Boiler | X | X | X | - |
| Wood Pellets Boiler | X | X | X | - |
| Power-to-Heat | X | X | X | - |
| Solar Flat Collector | X | X | X | - |
| CHP Gas micro | X | X | X | X |
| CHP Gas mini | X | X | X | X |
| Heat Pump Air | - | X | X | - |
| Low Temperature Oil Boiler | - | X | X | - |
| Photovoltaics | - | - | - | X |

As seen in the table, there are some technologies like a heat pump or a low-temperature oil boiler that can only provide heat at low temperatures or hot water level. For cases like these, a secondary technology can be used to heat up the output to a higher temperature level.

### 4.3. Time-Step Selection

There are two main assumptions necessary when it comes to the choice of time-steps in an optimization. Firstly, the length of each individual time-step has to be defined. Bottom-up energy models have a techno-economic focus and, hence, a relatively detailed time division is necessary to keep the logic of the model as realistic as possible. Furthermore, the time-step length must be sufficient to allow an appropriate simulation of the operation of technologies. Nevertheless, the disaggregation of the time horizon into shorter time-steps increases the run time of the optimization. Consequently, the time-step length is defined as one hour. This setting allows a relatively detailed modeling of reality and is a level that can still be optimized within a reasonable run time.

The second parameter is the total amount of time-steps considered in the optimization. On one hand, small number of time-steps can lead to a distorted result, which is not representative of the entire optimization period. On the other hand, a large number of time-steps can increase the run times enormously.

To minimize this distortion factor for the calculation, all time-steps are considered once in the entire optimization time horizon, however, not for every year in order to keep the optimization problem to a solvable size. The year is divided into four equal sections and a time rolling horizon is used for the complete optimization period. Figure 4 illustrates the selected sections for each of the expansion years. As can be seen, in the first year, the blue sections of time-steps are used, in the second year, the light grey section, and so on.

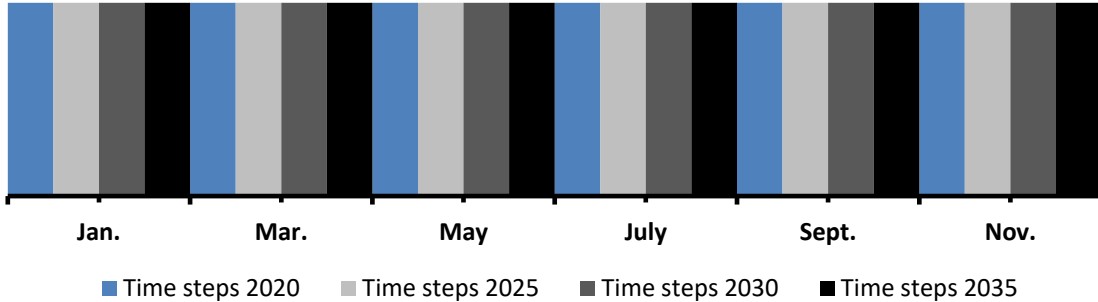

**Figure 4.** Time-step selection for the optimization years (2020, 2025, 2030, and 2035).

Every section includes 2190 time-steps and is used for one year of the optimization. The optimization has a time horizon of 15 years in which both expansion methods optimize. The model optimizes every fifth year including 2020, 2025, 2030, and 2035. As a result, a total number of 8760 time-steps are considered in the optimization.

The myopic expansion approach optimizes every year separately (2190 per period/year). Therefore, the method runs the optimization four times and only considers the information of the particular year. Furthermore, both expansion approaches can decide whether to install new capacities at the beginning of every year.

### 4.4. Discounting of Cost

For the perfect foresight approach, costs are discounted to account for inflation. In DISTRICT, costs are discounted within the model with an assumed discount factor. For the perfect foresight approach, the considered time steps are assigned to the corresponding optimization years in use. This allows discounting the future cost. For the myopic approach, an endogenous discounting does not influence the model decisions since each year is optimized separately and the same discount factor applies to all costs within one year. However, to make the results comparable, for this research, the costs are discounted within the optimization with the same factor as in the perfect foresight approach. For the present paper, a discount factor of 2% is assumed.

### 4.5. Energy Prices

Three primary energy types including oil, gas, and wood pellets are considered. All costs are based on the year 2015, the gas price being 6.8 ct/kWh [23]. In addition, taxes and grid fees (3 ct/kWh) are subtracted [24]. The subtraction is done because neither taxes nor grid fees are included in the electricity price. The resulting 3.8 ct/kWh are multiplied with 1.1 in order to get the costs of the facilities primary energy usage according to the calorific value [25]. Therefore, the primary costs for gas are 4.2 ct/kWh.

The cost of oil (8.0 ct/kWh) is derived from the Bundesminesterium für Wirtschaft und Energie (BMWi) [23]. The costs are set higher than stated in the report due to the massive price drop in recent years which leads to the expectation of prices rising again in the next years. Hence, to avoid biased results due to a temporary fall in prices, the oil costs are set to a higher level. The price for wood pellets is set at 4.7 ct/kWh and equals the price for a 5 tons delivery in Germany [26]. To map the future price increase, these prices for the primary energies are multiplied with a factor for every year to retrieve the corresponding prices for the optimization years, given in Table 3.

Another energy type that is considered in the optimization is electricity. In contrast to the considered fossil fuels, the electricity price varies for each time-step and represents the price per kWh at the spot market. The main characteristics of used spot market price projection can be found in the Appendix A in Table 3. The electricity procured at the spot market and imported into the system is accompanied by the corresponding $CO_2$ emissions derived from the power plants in use. Except for one scenario (VAR_CO2), this emission factor is constant for each year to limit the run-time.

**Table 3.** Emission factors and primary Energy Prices for each optimization year.

| Primary Energy | Emission Factor [g/kWh] | Price in 2020 [€/kWh] | Price in 2025 [€/kWh] | Price in 2030 [€/kWh] | Price in 2035 [€/kWh] |
|---|---|---|---|---|---|
| Oil | 267.8 | 0.084 | 0.088 | 0.092 | 0.097 |
| Gas | 198.5 | 0.049 | 0.057 | 0.066 | 0.076 |
| Wood pellets | 10.4 | 0.049 | 0.051 | 0.054 | 0.057 |

*4.6. Scenario Design*

The scenario selection is based on variations that represent a variety of different cases; see Table 4. The main goal is to show how different factors impact the outcome of both expansion methods. To achieve this, the scenarios vary in the technologies available for expansion and the manner in which $CO_2$ emissions are considered; compare Table 4. The $CO_2$ emission cost and average spot market price for the imported electricity are shown in Table 3 in the Appendix A.

**Table 4.** Scenario selection and main assumptions.

| Options | TECH | VAR_CO2 | EEX_SHOCK | RETRO |
|---|---|---|---|---|
| Tech expansion | x | x | x | x |
| Elec grid expansion | x | x | x | |
| Refurbishment | | | | x |
| spot market $CO_2$ emission | Constant | Variable | Constant | Constant |

4.6.1. Technology Scenario

First, a technology expansion (TECH) scenario is defined. This case allows an evaluation of the influence of technology expansion, i.e., investment cost and an almost continuous increase in outer boundary conditions like the spot market price and $CO_2$ emission costs. The TECH scenario has very basic expansion opportunities. It is capable of installing new generation and storage technologies within the buildings or in the central region. In addition, investments in grid expansion are possible. A moderate increase in the electricity price, as well as $CO_2$ emission cost, is assumed; see Table 3. In this case, a constant $CO_2$ emission encumbrance in the electricity mix is defined. This means that imported electricity is charged with a constant $CO_2$ emission value for each time-step.

4.6.2. Variable $CO_2$ Emission Scenario

The VAR_CO2 scenario is an extended scenario of the TECH scenario that additionally considers a variable $CO_2$ emission encumbrance within the subordinate electricity mix, which is supplied to the district via the spot market. Since the $CO_2$ emissions are charged with a penalty based on the $CO_2$ price, the objective of the system would be to avoid emissions as much as possible and hence the effect on the technology deployment and operation. This allows a more detailed evaluation of the operation and the differences in the expansion approaches since all other scenarios have a constant emission value. The developments of the emissions are illustrated in Figure 5. Both assumptions reflect the current emission reductions of the German power plant fleet, because of the increasing share of renewable energy and other decarbonization measures [27].

The electricity price development, as well as the expansion options, are however the same as in the TECH scenario.

4.6.3. EEX Price Shock Scenario

Since one of the main differences between the two expansion approaches is the duration of the foresight given to the solver, a scenario with a sudden increase in a price signal is included. In the EEX_SHOCK scenario, a peak in the electricity price is embedded in the second expansion year (2025), which is then reduced back to the reference value in the

third expansion year (2030) (see Figure 6) i.e., the EEX_SHOCK scenario increases the price, just for a short period of time, to a level that would normally occur in later periods.

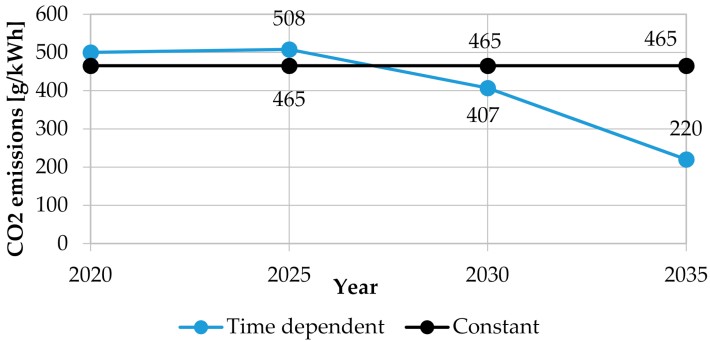

**Figure 5.** Average $CO_2$ emission factor in the electricity mix.

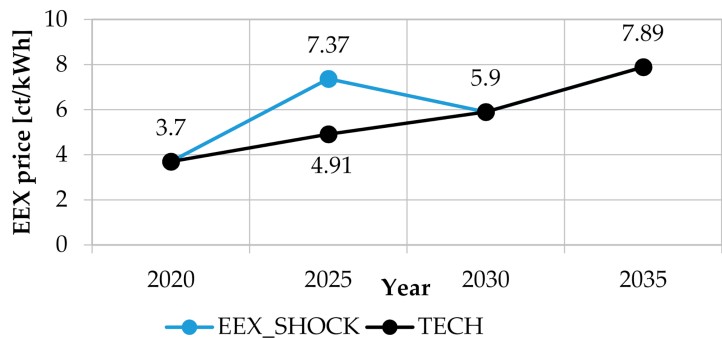

**Figure 6.** Average day-ahead spot market price for TECH and EEX_SHOCK scenarios.

The remaining parameters are left unchanged to the TECH scenario. This procedure makes it possible to analyze how the expansion approaches react to an extreme temporary change that deviates highly from the rather linear increase rate in the TECH scenario. Another aspect which is integrated into the scenario of the EEX_SHOCK is the non-negativity of the electricity prices as of the year 2025. Since 2008, negative day-ahead prices are allowed in the Spot Market. This price constellation usually occurs if the share of renewable energy generation is high and other energy plants are not reducing their production, due to inflexibility [28]. The price shocks permit these negative prices and, hence, the system is not able to make a profit from buying electricity. The cost development curves are presented in Figure 6.

### 4.6.4. High Initial Investment Scenario

The last scenario RETRO is chosen to represent a case with a high initial investment choice at the beginning of the optimization. Due to the methodology of the modeling approach of refurbishment in DISTRICT, the investment decision of building refurbishment can only take place in the first expansion year (2020). There are three retrofit levels implemented in the system: KfW100, KfW85, and KfW55. With the implementation of each level, the heat demand will change to low temperature, and the total heat demand decreases. The amount by which the heat demand is reduced with each building standard is illustrated in Figure 7. The figure states the cumulated heat demand of the whole system if each building has the same retrofit standard. The corresponding costs of the retrofit measures for each building are listed in the Appendix A (Table A2). Since building refurbishment represents a very large investment, the different expansion approaches could potentially lead to different results in the refurbishment decision.

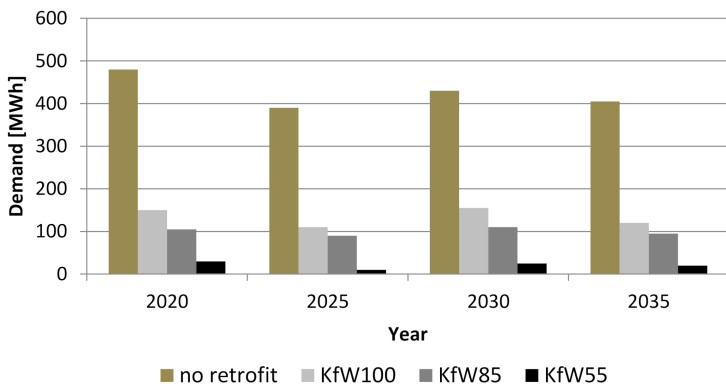

**Figure 7.** Annual heat demand for the regarded system, assuming the entire building stock has the same building standard.

Like in the TECH scenario, the *RETRO* scenario has a moderate increase in the spot market price development over the optimization period. However, the *RETRO* scenario considers in one case a higher rise in the $CO_2$ prices over time, and a moderate one in a second case.

## 5. Results

In this chapter, the scenario results are presented and discussed. The first section presents the results of each scenario which is then followed by a general discussion of the results.

### 5.1. Technology Scenario

The TECH scenario, with the possibility to invest into generation technologies as well as grid capacity, shows that the objective values of the two expansion methods are almost identical. However, the grid expansion decisions of the perfect foresight and myopic approach differ.

Both of the expansion approaches only invest in grid capacities in the first optimization year (2020), but the myopic approach installs more connection and transformer capacities than the perfect foresight approach. The difference between the investment decisions for new grid capacities is given in Table 5. Here, and in the remainder of the paper, all differences are defined as value = [(result perfect foresight—result myopic)/(result perfect foresight)]

**Table 5.** Comparison of expanded grid capacities in the first optimization year of the TECH scenario.

| Scenario | Grid Component | Perfect Foresight [kW] | Myopic [kW] | Difference to Myopic |
|----------|---------------|------------------------|-------------|----------------------|
|  | Transmission Line | 540 | 820 | −51.9% |
| TECH | Transformer | 290 | 448 | −54.5% |
|  | Generation capacities | 1471 | 1634 | −101.1% |

The different grid and generation capacity decisions also affect the resulting cost structure. The additional capacities of the myopic approach lead to an increase in the investment and fixed costs in comparison to the perfect foresight approach as seen in Table 6.

Moreover, the spot market costs are also higher in the myopic approach. In the myopic approach more investments in power-to-heat technologies are undertaken than in the perfect foresight approach; see Table 4. This leads to a higher electricity demand, increasing the amount of electricity procured at the spot market. In the long run, however, the different technology mix leads to lower variable costs. Hence, the variable and fuel costs are higher in the perfect foresight approach, because of the need to compensate for the missing heating

energy from power to heat with alternative technologies such as gas boilers. Overall, the higher fixed and investment cost outweigh the lower variable cost so that regarding the total cost; both approaches reach the same level. However, the development of the results differs over time between the approaches; compare Figure 8. Although the objective values converge in the target year, based on differing investment decisions there is a significant difference between the development of investment and spot market cost. It can be seen that the difference decreases over time due to the aforementioned investment decisions.

**Table 6.** Comparison of objective value and costs of the TECH scenario.

|  | Cost | Perfect Foresight [Euro] | Myopic [Euro] | Difference to Myopic |
|---|---|---|---|---|
| | Objective value | 185,217.15 | 185,432.67 | −0.12% |
| | Investment cost | 21,559.50 | 22,138.20 | −2.68% |
| TECH | Fix cost | 13,637.70 | 14,021.10 | −2.81% |
| | Variable cost | 59,920.10 | 56,463.60 | 5.77% |
| | Spot market cost | 65,389.70 | 67,165.30 | −2.72% |
| | Emission Cost | 25,154.67 | 25,660.00 | −2.01% |

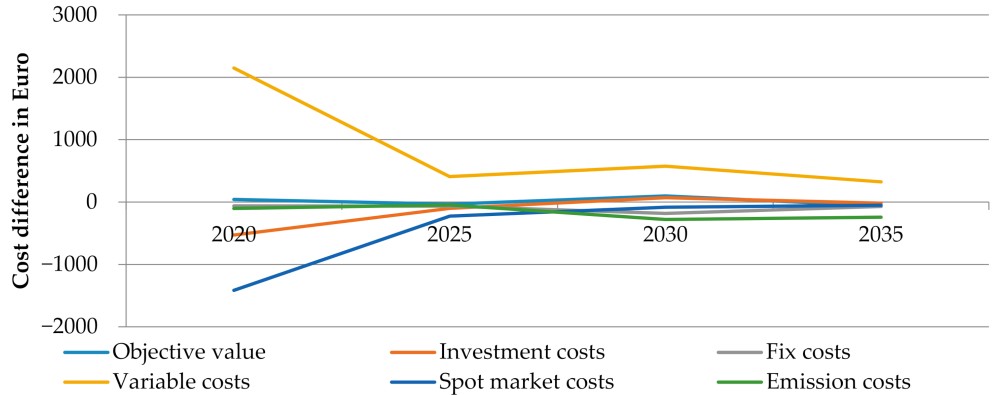

**Figure 8.** Cost difference between the two approaches over the optimization period.

### 5.2. Variable $CO_2$ Scenario

In the CO2_VAR scenario, the objective values of both approaches do not differ considerably. The main difference is found in the investment cost; see Table 7. In the perfect foresight approach, 30% less connection and transmission capacities are deployed in comparison to the myopic approach. A similar trend is observed in the generation capacities. The myopic approach deploys around 10% more generation technologies. These two decisions lead to the higher investment cost in the myopic approach. Due to the additional deployed technologies and especially the grid expansion, the fix costs are also respectively higher in the myopic approach.

**Table 7.** Comparison of objective value and costs of the CO2_VAR scenario.

|  | Cost | Perfect Foresight [Euro] | Myopic [Euro] | Difference to Myopic |
|---|---|---|---|---|
| | Objective value | 180,651.1 | 181,512.8 | −0.5% |
| | Investment cost | 18,868.0 | 22,260.3 | −18.0% |
| CO2_VAR | Fix cost | 12,361.9 | 13,945.2 | −12.8% |
| | Variable cost | 56,628.6 | 55,132.2 | 2.6% |
| | Spot market cost | 70,875.8 | 68,415.2 | 3.5% |
| | Emission Cost | 5431.33 | 5952.32 | 9.6% |

Another difference that can be observed is the time of deployment. In the perfect foresight approach, wood pellet boilers are installed earlier than the myopic approach, whereas PV deployment takes place at the last expansion year; compare Table 4. The

resulting installed capacities at the end of the optimization is, however, the same in both approaches; see Table 4. Another difference in the presented scenario is a deployment of battery storage systems in the myopic approach. The investment decision is based on the opportunity of importing electricity at low prices and low emission costs. This however is only valid in the early years, and in the later periods, this benefit is reduced significantly. Hence, in the perfect foresight approach, this investment is not worthwhile based on the knowledge of the upcoming years. Thus, the system knows that charging costs will increase considerably. Since the information of increased electricity prices and higher emission costs are not known to the myopic approach, the battery storage systems are considered a useful investment.

### 5.3. EEX Price Shock Scenario

The objective values are almost identical in each case, as in the previous scenarios. The values are presented in Table 8.

**Table 8.** Comparison of objective value and costs of the EEX_SHOCK scenario.

|  | Cost | Perfect Foresight [Euro] | Myopic [Euro] | Difference to Myopic |
|---|---|---|---|---|
|  | Objective value | 192,889.60 | 193,510.10 | −0.3% |
|  | Invest cost | 21,016.10 | 26,532.40 | −26.2% |
| EEX_SHOCK | Fix cost | 13,599.80 | 14,105.40 | −3.7% |
|  | Variable cost | 65,474.20 | 61,729.00 | 5.7% |
|  | Spot market cost | 68,281.30 | 64,825.90 | 5.1% |
|  | Emission Cost | 6356.00 | 6583.08 | 3.6% |

However, the grid expansion decisions of the perfect foresight and myopic approach differ. In both expansion approaches, investments into grid infrastructure are undertaken in the first optimization year (2020) only, but the myopic approach installs higher connection and transformer capacities than the perfect foresight approach; compare Table 9.

**Table 9.** Comparison of newly installed grid components and generation capacities of the EEX_SHOCK scenario.

| Scenario | Grid Component | Perfect Foresight [kW] | Myopic [kW] | Difference to Myopic |
|---|---|---|---|---|
|  | Transmission Line | 490 | 820 | −67.3% |
| EEX_SHOCK | Transformer | 254 | 448 | −76.4% |
|  | Generation capacities | 1482 | 1715 | −15.7% |

The table shows that the difference between the expansion approaches is higher than 67% for connection and transformer capacities, respectively. The additional grid capacities allow the myopic approach to using a higher amount of power-to-heat, which increases the electricity demand and thus the amount of electricity sourced at the spot market since there is no local electricity generation in the first year; see Figure 9. In the last optimization year, the difference in the installed capacities reaches 16%.

Another difference in the installed technologies is the time of the PV deployment. In the myopic approach, the PV deployment takes place in the second expansion year (2025), wherein the perfect foresight the deployment takes place in the following expansion year (2030). This is due to the fact that in the perfect foresight approach, the development of spot market prices as well as the decrease in PV investment cost are known, whereas the myopic approach does not have this information and hence starts installing PV as soon as it becomes profitable compared to procuring electricity at the spot market. The opposite effect can be observed in the wood pellet boiler deployment. In this case, the boilers are deployed to a higher extent in early stages in the perfect foresight approach. The reasons are the upcoming increase in the $CO_2$ and electricity prices in the following years.

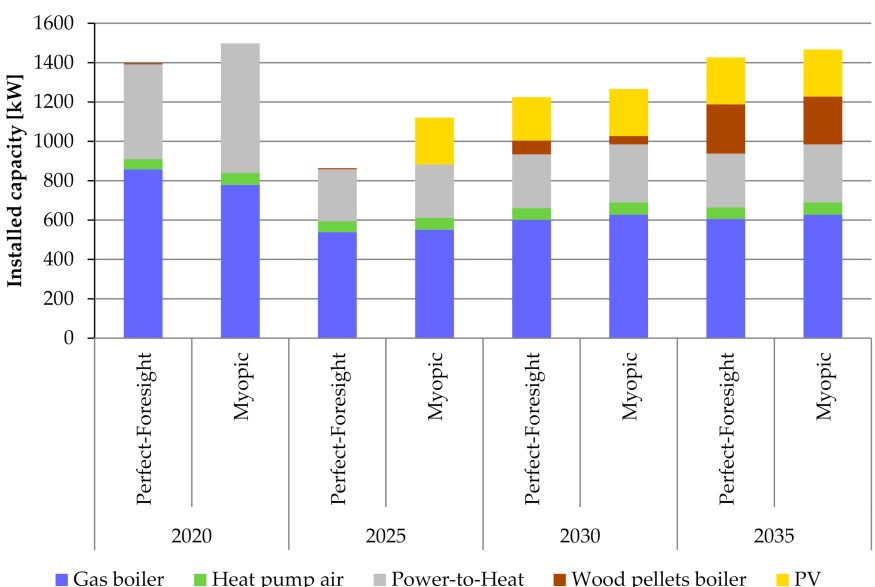

**Figure 9.** Installed generation capacities in the EEX_SHOCK scenario.

The total additional capacities of the myopic approach lead to higher investment and fixed costs in comparison to the perfect foresight approach. In addition, the variable and fuel costs are higher for the perfect foresight approach, as the missing heating energy from power to heat has to be compensated with alternative technologies such as gas boilers with their corresponding fuel and variable costs. The trade costs in the *EEX_SHOCK* scenario are lower in the myopic approach, due to the previously mentioned earlier installation of PV power plants and the resulting reduction of imported electricity from the spot market.

However, the increased and earlier PV installations of the myopic approach do not lead to noticeably lower $CO_2$ emissions. The increased power-to-heat technologies require a large amount of electricity, which is imported from the spot market and therefore charged with $CO_2$ emissions. This additional electricity demand and thus import nearly outweighs the reduction in emission achieved by higher PV capacities. Effectively, the $CO_2$ emissions are lower in the myopic scenario, but as considerably as one could have expected when looking at the installed PV capacities.

### 5.4. High Initial Investment Scenario

Building retrofit requires a high upfront investment, which only becomes beneficial in the long run. The profitability is obtained by reducing the heat demand, which leads to corresponding savings by reduced operating costs and cheaper reinvestments when a technology needs to be replaced.

The results show that the objective values of the expansion approaches are almost identical for both cases. However, with the perfect foresight approach, a noticeable difference in the investment and variable costs is observed; see Table 10. This is due to the retrofit investment decisions. Compared to the myopic approach, the model opts to refurbish one additional building in the perfect foresight approach.

**Table 10.** Objective and cost values of the RETRO cases.

|  | Cost | Perfect Foresight [Euro] | Myopic [Euro] | Difference to Myopic |
|---|---|---|---|---|
|  | Objective value | 176,046.36 | 176,512.71 | −0.26% |
|  | Invest Cost | 75,819.38 | 67,947.28 | 10.38% |
|  | Fix Cost | 13,324.42 | 13,831.63 | −3.81% |
| RETRO | Variable Cost | 15,442.78 | 20,983.26 | −35.88% |
|  | Spot market Cost | 54,415.46 | 55,641.64 | −2.25% |
|  | Emission Cost | 17,044.33 | 18,108.89 | −6.25% |

Figure 10 illustrates the difference between the retrofit decisions of the perfect foresight and myopic approach. This leads to a reduced heat demand in the perfect foresight approach, which is accompanied by higher invest, but considerably lower variable costs.

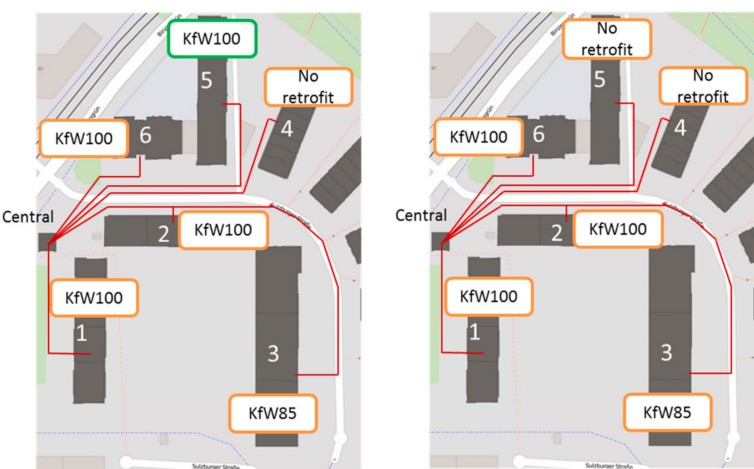

**Figure 10. Left** picture displays retrofit decisions of the perfect foresight and the **right** picture of the myopic approach.

This leads to a difference in the heat demands of the expansion approaches, whereby the decrease over the optimization period in the perfect foresight approach is 28%, which corresponds to 159 MWh. The additional retrofit also has an impact on the cost structures of the expansion approaches. On the one hand, the retrofit leads to a high increase in investment costs for the perfect foresight approach. On the other hand, operational costs, including fuel and emission costs, are significantly lower in comparison to the myopic approach. Figure 11 shows the heat generation for the two approaches, which confirms the fact that in total, the perfect foresight approach achieves lower operation and emission costs due to the reduction in heat demand and hence in necessary heat generation.

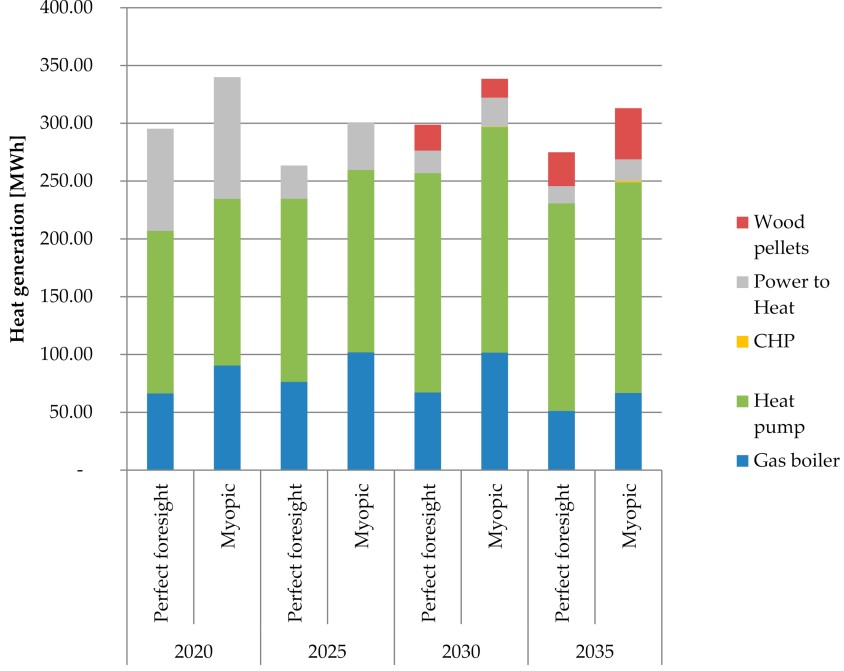

**Figure 11.** Heat generation in the RETRO scenario for the two expansion approaches.

The RETRO scenario shows an increased amount of installed generation capacities for the myopic approach. Moreover, the difference is even more crucial in cost developments with an extra retrofit of building five. This is caused by the decrease of needed heat demand and, therefore, less need for generation capacities for heating. The myopic method installs 2.2% more generation capacities, which are shown in Figure 11, with the reference $CO_2$ price increase and 6.1% more when a high $CO_2$ price increase is considered. The major effect of the additional retrofit investment decision is the decreasing $CO_2$ emissions of the energy system. This amount is reduced by 35 tons due to the lower heat demand for the perfect foresight approach. This amounts to 6.4% lower emissions in this case.

The cost development over time (Figure 12) shows that the driver for the difference of the objective value is the investment decision in the first year. Although the relative difference is fairly small with 0.1%, it is obvious that with such costly investment decisions, these become the driver for the future development. It can also be seen that the mitigation of rising $CO_2$ emissions in the myopic scenario by increasing emission costs as well as necessary investments into generation technologies outweigh the savings from not retrofitting in the beginning.

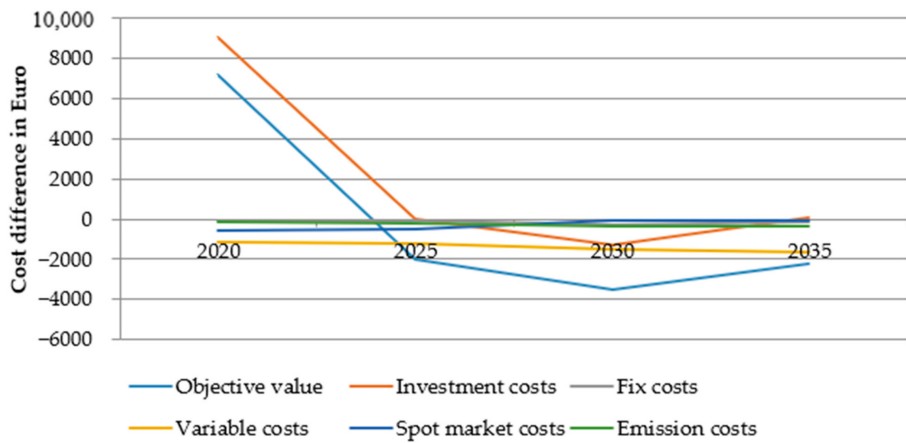

**Figure 12.** Cost difference between the two approaches over the optimization period for the RETRO scenario with high $CO_2$ prices.

### 5.5. Discussion

Figure 13 summarizes the relative difference between the cost elements of the objective value and the latter itself for all scenarios. The paper cannot completely confirm the preference of conventional generation technologies in the myopic approach found by [3], but there is a preference for a postponement of investments, especially pronounced in the RETRO scenario. As many of the considered technologies can be regarded as advanced technologies with high investment cost, this is in line with the findings of [18]. A previous study [18] showed that "the shorter the foresight, the later the adoption of an advanced, but currently expensive, technology". One explanation for this effect may well be the maximization of short term profits by the myopic approach, which is also observed by [13].

The most significant differences between costs can be observed for those scenarios with either a shock-like development or when high investment options to avoid $CO_2$ emissions are combined with more strongly increasing $CO_2$ costs so that avoiding $CO_2$ emissions becomes a necessity. This is either achieved by a higher retrofit rate, as in the perfect foresight approach or by mitigating the effects of higher $CO_2$ emissions in later years as in the myopic approach. Without considerable price increases for $CO_2$ emission certificates, the costs are neither sufficient to justify building refurbishment nor do they impose a need to mitigate increasing $CO_2$ emissions by any means.

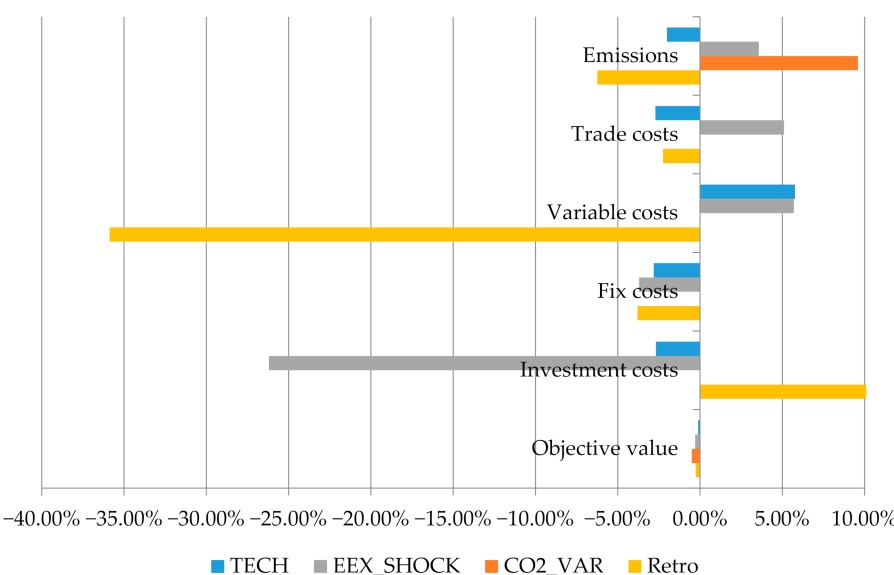

**Figure 13.** The relative difference between the results of the myopic and the perfect foresight approach in each scenario, the difference is displayed as value = [result perfect foresight − result myopic]/result perfect foresight.

For enforced restrictions, e.g., limits to GHG emissions, [1] proposes an investigation with a combined application of both approaches, as he finds that "the resulting costs of lost opportunities can be of the same order of magnitude as the mitigation costs themselves". This seems also fitting for scenarios that include retrofit, as it has very high upfront costs combined with a large potential to save GHG emissions. The presented scenarios with DISTRICT include $CO_2$-prices rather than limits to GHG emissions so that the cost of mitigating the additional emissions in the myopic scenario add up to a comparable objective value rather than choosing retrofit in the beginning, supporting the findings from [1]. Nevertheless, it also shows that investment options with high upfront costs need longer planning horizons, making them less attractive than other options, which is arguably one of the main reasons retrofit is not realized at the desired pace. Including retrofit endogenously is also one of the main differences to national energy system models, where these decisions are usually external factors and less detailed.

As can be seen in Figure 14, the optimization runs with the perfect foresight approach take significantly longer than the myopic approach. As optimization complexity and the thereby resulting run-times are often an issue among researchers, it can be concluded that the myopic approach reduces complexity and thus run-time significantly, as was the case in [19]. This then allows researchers to assess more complex options within the model that might not even be solvable with the perfect foresight approach or may require run-times that exceed several weeks. Depending on the conditions in the researchers work environment, the latter might not only not be desired, but also impossible to wait for.

In conclusion, the myopic approach prefers to invest into generation technologies with low variable costs, as they are cheaper in the short-term. Investment into more efficient, but expensive technologies would require a rise in external prices or carbon emission costs. This takes place in later years, but the model cannot see this when making first investment decisions. Accordingly, the myopic approach shows less investment into demand reduction technologies (retrofit) as investment is very high and the positive effects become relevant in later optimization years. This creates lock-in effects in later optimization years, so that with stable parameters and options for retrofit, the myopic approach shows higher emission costs. These effects allow the total costs converge, leading to more or less the same objective values. This has to be considered while interpreting the results. Using DISTRICT with the myopic expansion approach allows for the generation of consistent results with a considerably shorter run-time under the condition that input parameters are stable,

similar to the results of [1,19] for their corresponding models. Shock-like events, suddenly enforced restrictions, or options with very high upfront investment costs like building retrofit, however, generate very different results. Thus, it cannot be assumed that the myopic approach is valid when analyzing refurbishment rates or similar costly investments that lead to considerable reductions in GHG emissions. If the researcher wants to find the optimal mix between refurbishment and renewable energy generation, the perfect foresight approach is still recommended. If due to other restrictions the myopic approach has to be used, the result should be analyzed regarding the cost spend on $CO_2$ emission certificates and then compared to the alternative of retrofitting additional buildings. Otherwise, the result might not depict the complete picture.

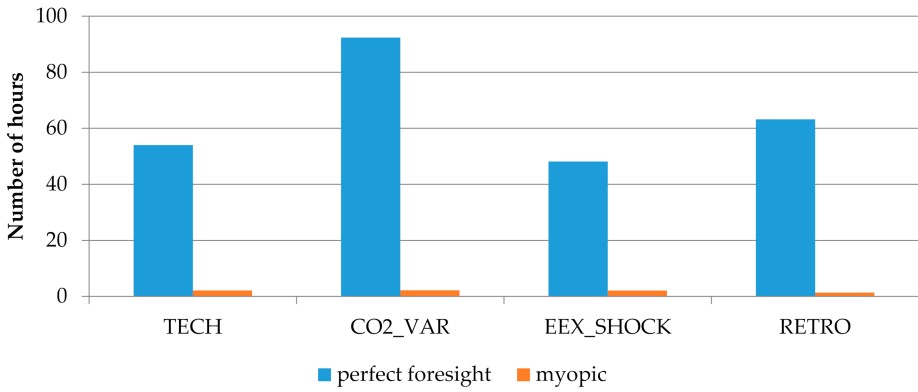

**Figure 14.** Duration of the optimization runs for the complete time horizon in hours.

Knowing in which direction the myopic approach shifts the results (towards higher investment and fix costs and lower variable costs for generation technologies and less invest into demand reduction, while reaching fairly similar total cost), enables researchers to benefit from the shorter run-times while maintaining reasonable results. Taking into account that the end systems are quite similar for the two approaches, the myopic approach is considered a good fit when optimizing energy systems with a very high endogenous complexity. For the DISTRICT model, especially the different temperature levels and expansion options for retrofit, heat and electricity grid increase the complexity enormously. Reducing the run-times with the myopic approach allows increasing the technological, as well as time and spatial resolution (e.g., including more buildings). The appropriateness of the myopic approach for certain scenario frameworks is in line with the findings of [1,12,19].

## 6. Conclusions

In this paper, the impact of applying a myopic and perfect foresight approach to the regional energy system optimization model DISTRICT is analyzed.

The analysis at hand shows that for regional scenario layouts with stable input parameters, both approaches lead to a very similar result with marginal differences. In scenario settings with shock-like events, the results in the year with the shock differ considerably. If researchers desire to analyze the shock as an unforeseen event, only the myopic approach can enable this. In scenario settings that include technology options with very high upfront costs such as building refurbishment, the refurbishment decisions differ, leading to different energy systems in the target year. However, the savings from realizing that less refurbishments are needed to mitigate additional $CO_2$-emissions in later years so that the total cost is comparable at the end of the optimization period. This indicates that in order to investigate the optimal system layout including very costly, long-term investments, the myopic approach is not a perfect fit.

Generally, with the perfect foresight approach there is a tendency towards higher investment and fix costs compared to the myopic optimization. This is due to the fact that technologies with lower operating costs are preferred by the myopic approach, leading to higher import and mitigating costs in later years, so that over the whole optimization period,

both approaches reach similar objective values. Bearing this in mind when interpreting results, the myopic approach allows reducing run-times by more than 90%. This allows the inclusion of more time-steps, technologies, or model areas compared to the perfect foresight approach, allowing a higher level of detail. Especially in local energy systems, this means one could include larger areas when modeling at building level, including more technologies and sectors, such as cooling grids, Power to X, or electro mobility, without risking the model to reach an infeasible problem-size. Additionally, this means more scenarios can be run and analyzed at the same time. In future work, these technologies can be further analyzed with the interaction on the modeling approach such as perfect foresight or myopic optimization as well as with stochastic approaches. It can be also tested if one approach is more suitable on reaching climate targets when considering the diverse stakeholders at the local level. Different stakeholders have different objectives and knowledge levels, which might correspond better with one approach or the other. This may indicate if specific technologies or transformation pathways have to be adapted or intensified.

**Author Contributions:** Conceptualization, J.T. and N.S.H.; Data curation, A.D.; Investigation, N.S.H. and A.D.; Methodology, J.T. and N.S.H.; Supervision, C.K.; Validation, J.T. and A.D.; Writing—original draft, J.T. and N.S.H. All authors have read and agreed to the published version of the manuscript.

**Funding:** This research was funded by the German Ministry for Economic Affairs and Energy within the projects "NEMAR" project 03ET4018A) and "STROWAE" (project 0325736).

**Acknowledgments:** The publication of this work has been financially supported by the German Ministry for Economic Affairs and Energy within the project "NEMAR" project 03ET4018A) looking at the electricity and smart grid infrastructure and the project "STROWAE" (project 0325736) analyzing the coupling of the heat and electricity sector.

**Conflicts of Interest:** The authors declare no conflict of interest.

## Appendix A

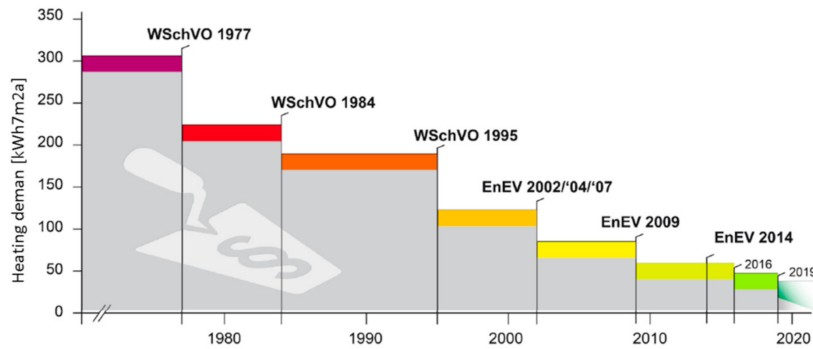

**Figure A1.** EnEV standards according to [29].

The Kreditanstalt für Wiederaufbau (KfW) standards are compared to the standards from the German energy efficiency law (EnEV) standard. The main criterion is the reduction in demand. KfW100 is equivalent to the current EnEV standards, where KfW85 requires a reduction of 15% reaching a total demand of 85% of the EnEv standard. KfW55 requires a 45% reduction, respectively.

**Table A1.** Costs of the generation technologies [30].

| Energy Generation Technologies | Invest Costs [€/kW] | Fix Costs [€/kW_a] | Variable Costs [€/kWh] |
|---|---|---|---|
| Low temperature boiler | 350 | 7.8 | 0.0004 |

**Table A1.** *Cont.*

| Energy Generation Technologies | Invest Costs [€/kW] | Fix Costs [€/kW_a] | Variable Costs [€/kWh] |
|---|---|---|---|
| Gas boiler | 175 | 4.38 | - |
| CHP Gas micro | 1700 | 17 | - |
| CHP Gas mini | 2070 | 41.4 | - |
| Wood Pellets | 788 | 8.3 | 0.0004 |
| Power to Heat | 50 | 1.5 | - |
| Heatpump air | 1195 | 29.88 | - |
| Solar Flat Collector | 595 | 8.93 | - |
| Photovoltaics | 975 (year 2020) | 30 | - |

**Table A2.** Refurbishment cost for the different building standards.

| Building/Region | KfW100 [€] | KfW85 [€] | KfW55 [€] |
|---|---|---|---|
| 1 | 192,861 | 216,608 | 428,017 |
| 2 | 168,816 | 189,204 | 376,390 |
| 3 | 421,376 | 472,083 | 1147,776 |
| 4 | 106,934 | 118,643 | 198,153 |
| 5 | 188,767 | 219,934 | 456,350 |
| 6 | 144,043 | 167,778 | 373,552 |

**Table 3.** $CO_2$ emission cost and average spot market prices for the calculated scenarios.

| | | 2020 | 2025 | 2030 | 2035 |
|---|---|---|---|---|---|
| | TECH | 10 | 25 | 50 | 85 |
| $CO_2$ emission cost Euro/t | VAR_CO2 | 10 | 25 | 50 | 85 |
| | EEX_SHOCK | 10 | 25 | 50 | 85 |
| | RETRO | 10 | 25 | 85 | 125 |
| | TECH | 3.7 | 4.91 | 5.9 | 7.89 |
| Average annual spot market price in Euroct/kWh | VAR_CO2 | 3.7 | 4.91 | 5.9 | 7.89 |
| | EEX_SHOCK | 3.7 | 7.37 | 5.9 | 7.89 |
| | RETRO | 3.7 | 4.91 | 5.9 | 7.89 |

**Table 4.** Installed generation capacities in all scenarios.

| | TECH | | | | | | | |
|---|---|---|---|---|---|---|---|---|
| | 2020 | | 2025 | | 2030 | | 2035 | |
| | Perfect foresight | Myopic | Perfect foresight | Myopic | Perfect foresight | Myopic | Perfect foresight | Myopic |
| Gas boiler | 752 | 752 | 467 | 484 | 243 | 260 | 258 | 263 |
| Heat pump air | 86 | 83 | 96 | 97 | 127 | 140 | 127 | 140 |
| Oil boiler | 0 | 1 | 13 | 1 | 18 | 1 | 18 | 1 |
| Power to heat | 245 | 254 | 259 | 267 | 264 | 275 | 264 | 275 |
| PV | 0 | 0 | 0 | 0 | 217 | 239 | 239 | 239 |
| Thermal storage | 2 | 4 | 8 | 6 | 12 | 11 | 21 | 16 |
| Wood pellets boiler | 0 | 0 | 0 | 0 | 19 | 13 | 39 | 48 |

**Table 4.** *Cont.*

| | EEX_Shock | | | | | | | |
|---|---|---|---|---|---|---|---|---|
| | 2020 | | 2025 | | 2030 | | 2035 | |
| | Perfect foresight | Myopic | Perfect foresight | Myopic | Perfect foresight | Myopic | Perfect foresight | Myopic |
| Gas boiler | 858 | 779 | 656 | 710 | 456 | 533 | 456 | 533 |
| Heat pump air | 52 | 60 | 53 | 60 | 59 | 64 | 59 | 64 |
| Power-to-Heat | 481 | 659 | 481 | 659 | 482 | 659 | 482 | 659 |
| Wood pellets boiler | | | | | 36 | 8 | 246 | 220 |
| PV | | | | 239 | 221 | 239 | 239 | 239 |
| | CO2_VAR | | | | | | | |
| | 2020 | | 2025 | | 2030 | | 2035 | |
| | Perfect foresight | Myopic | Perfect foresight | Myopic | Perfect foresight | Myopic | Perfect foresight | Myopic |
| Gas boiler | 831 | 777 | 597 | 630 | 397 | 479 | 605 | 627 |
| Heat pump air | 55 | 60 | 59 | 62 | 64 | 64 | 64 | 65 |
| Power-to-Heat | 544 | 656 | 544 | 656 | 545 | 656 | 559 | 656 |
| Wood pellets boiler | | | | | 55 | 2 | 223 | 200 |
| PV | | | | | | 208 | 239 | 239 |
| | RETRO | | | | | | | |
| | 2020 | | 2025 | | 2030 | | 2035 | |
| | Perfect foresight | Myopic | Perfect foresight | Myopic | Perfect foresight | Myopic | Perfect foresight | Myopic |
| Gas boiler | 752 | 752 | 467 | 484 | 240 | 256 | 242 | 266 |
| Heat pump air | 84 | 87 | 97 | 97 | 131 | 140 | 131 | 140 |
| Oil boiler | 0 | 0 | 0 | 0 | 0 | 0 | 0 | 0 |
| Power to heat | 248 | 254 | 262 | 274 | 265 | 274 | 265 | 274 |
| PV | 0 | 0 | 0 | 0 | 218 | 239 | 239 | 239 |
| Thermal storage | 1 | 4 | 4 | 6 | 7 | 10 | 17 | 15 |
| Wood pellets boiler | 0 | 0 | 0 | 0 | 22 | 14 | 41 | 46 |

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
