# Peer review of "Effect of the Foresight Horizon on Computation Time and Results Using a Regional Energy Systems Optimization Model"

_energies, doi:10.3390/en14020495_

Round 1
Reviewer 1 Report
The paper deals with a comparative study on the impact of two foresight horizon approaches (myopic and perfect foresight) by using the regional energy system optimization model DISTRICT based on a case study of 6 buildings located in the Weingarten district (Freiburg, Germany). The paper describes very clear the proposed methodology and the obtained results are discussed in details.
The following issues are recommended to improve the paper:
- Introduction, lines 60-63: all the section numbers are zero instead of 2 to 5!
- Several typing mistakes should be solved, e.g. paragraphs generated artificially by splitting a sentence, different fonts used in the same text body, use “Table n” instead of “table n”, etc.
- Figure 2 should be revised, it is not clear the role of “Objective Value1” and “+” in the right side of the figure.
- Table 1: explain the significance of “MWh/a thermal”
- Lines 206-207: “Figure 4 illustrates the selected sections for each of the expansion years. As can be seen, in the first year, the green sections”. The colour used in Figure 4 for 2020 is blue.
- Results and discussions: generally, the myopic approach yields to higher investment into grid components and generation capacities comparing to the perfect foresight approach. Please explain why these differences happen, considering each approach fundamentals. A general statement can be useful in this discussions part (such explanation is given at the end of the paper, in the Conclusion section).
Reviewer 2 Report
The paper is interesting and the topic is timely. The structure is convincing and clear to the reader. It is easy to follow. But alas, there are few issues that must be addressed as follows
- The abstract does not communicate the research results well. So, please expand it and summarize the most impressive achieved result of the presented work in the abstract.
- The introduction section is short and does not motivate the reader. Please add a list of contributions to Sec 1. Use, please a bullet points list.
- Also, with each of the contributions, please mention the section that discusses each.
- The authors must add a new section dedicated to the literature review.
- In terms of the types of energy systems, the authors may need to expand this area in the paper and discuss the different types in order to highlight which type their work follow, for example https://ieeexplore.ieee.org/document/8737691
- How comes Figure 8 has values below zero, in minus? Is that realistic?
- What are the units used to measure the y-axis in figure 8?
- Add future work to the conclusion section?
- Please proofread the entire paper
Reviewer 3 Report
Introduction seems to be too short. Motivation of the paper is weak. Authors should present the gap in existing literature and propose alternative solutions.
Literature review is well prepared, however some newer papers (written in 2020) are recommended. Methodology, design of case study and results are well described. The section with conclusions is very short comparing with research and described simulations.
Author Response
We thank the reviewer for the time to read the article and the comments. Following, we have prepared our answers to the individual comments.
Introduction seems to be too short. Motivation of the paper is weak. Authors should present the gap in existing literature and propose alternative solutions.
- We added the following lines to explain the motivation and research gap more clearly:
28-30
39-40
63
135-139
Since the paper adds to the discussion about the suitability of different foresight approaches in sector-coupled optimization models we do not completely understand what is meant by adding alternative solutions. None of the cited publications have proposed alternative optimization methods but analyzed those existing ones for power sector models. We now do this for a sector-coupled model, including heat, cooling and electricity, and analyze whether this behaves similarly or whether a certain approach might be unfit to research questions with a focus on sector-coupling and / or related aspects.
Literature review is well prepared, however some newer papers (written in 2020) are recommended.
- We did an additional round of literature research but failed to find more recent publications than the ones included on this particular topic.
Methodology, design of case study and results are well described. The section with conclusions is very short comparing with research and described simulations.
- We prefer a concise conclusion that is based on extensive results and discussion in order to be comprehensible and do not see any advantage in making it longer than necessary.

Reviewer 4 Report
Very interesting article. In my opinion, it is ready to be published.
Author Response
We thank the reviewer for the comments and the time to read the article.